# Connexin 26 and Connexin 43 in Canine Mammary Carcinoma

**DOI:** 10.3390/vetsci6040101

**Published:** 2019-12-09

**Authors:** Savannah Luu, Cynthia Bell, Sarah Schneider, Thu Annelise Nguyen

**Affiliations:** Department of Diagnostic Medicine/Pathobiology College of Veterinary Medicine, Kansas State University 1800 Denison Avenue, Manhattan, KS 66506, USA; swluu@vet.k-state.edu (S.L.); dr.bell@sopforanimals.com (C.B.); smschneider@vet.k-state.edu (S.S.)

**Keywords:** canine mammary tumor, connexin 43, connexin 26, gap junctions, xenograft tumor

## Abstract

Incidence of canine mammary carcinoma is two times higher than the rate of human breast cancer. Mammary tumors are the most common type of cancer in intact female dogs and account for about half of all neoplasms in these dogs. Well-established models of breast cancer have shown that neoplastic cells often have a loss of intercellular communication, particularly gap junction proteins. Thus, the objective of this study is to explore the aspect of gap junction intercellular communication in canine mammary carcinoma, non-cancerous (CMEC) and cancerous (CMT12, CMT27, and CF41.Mg) cells, and patient-derived tumors. Both non-cancerous and cancerous mammary cells express connexins 26 and 43 using immunofluorescence; however, the level of expression is significantly different in quantitative analysis using western blot in which connexin 43 in both CMT12 and CMT27 is significantly decreased compared to CMEC. Furthermore, a decrease of gap junction capacity in CMT12 and CMT27 was observed compared to CMEC. Immunostaining of CMT27-xenograft tumors revealed positive Cx26 and negative Cx43 expression. Similarly, immunostaining of spontaneous canine mammary tumors revealed that Cx26 is present in all tumors while Cx43 is present in 25% of tumors. Overall, the study provides for the first time that a differential pattern of connexin expression exists between non-cancerous and cancerous mammary cells in dogs. This study will pave the path for further in vitro work of connexins in comparative canine models and possibly allow for novel therapeutics to be developed.

## 1. Introduction

Mammary cancer is the most common type of neoplasia in female dogs and accounted for 70% of all cancer cases in Genoa, Italy in 1985–2002 [1]. The incidence is higher when routine ovariohysterectomies/ovariectomies are not performed before two years of age and also in dogs six years of age and older, with a mean of 9–11 years of age [2]. Therefore, the incidence of canine mammary tumors varies throughout the world depending on the commonality of spaying. Furthermore, a retrospective study, focused on 11,544 dog biopsies from January 2002 to December 2012, demonstrated that mammary tumors in female dogs are a major health problem and approximately 50% are malignant [3].

In many cancers, a form of communication called gap junction intercellular communication (GJIC, Melbourne, Australia) is impaired [4,5]. Gap junctions are composed of two hemichannels, also known as connexons—one from each cell; each connexon is composed of a ring of six connexin (Cx) proteins that surround a hydrophilic pore. This structure allows a variety of molecules such as cAMP and other second messengers and inorganic molecules, like K^+^ and Ca^2+^, to pass between cells [6]. Connexin genes are known tumor suppressor genes in that they allow cells to effectively communicate and therefore differentiate; defects in connexin proteins can lead to decreased GJIC, thus leading to dysregulation of the cell cycle and eventually cancer initiation [7]. Gap junctions are different from other membrane channels because they are not selective to specific ions or molecules but instead allow molecules less than 1000 Daltons to pass from one cell to an adjacent cell; this ability of gap junctions makes them unique therapeutic targets [8]. Gap junctions are involved in critical functions such as cell homeostasis; therefore, conditions that decrease GJIC can be pathologic and life-threatening [5,9].

Connexins are named according to their molecular mass and gap junction hemichannels can be composed of one (homomeric) or more than one (heteromeric) type of connexin [10]. Different types of connexin are expressed as cells undergo differentiation; furthermore, the patterns of expression are cell- and tissue-specific [11]. In human breast cancer, it is known that GJIC capacity is significantly decreased due to decreased expression of connexin proteins or failure of connexins to assemble into functional gap junctions, especially Cx26 and Cx43 [4,12,13,14]. In normal human breast tissue, Cx43 is primarily expressed between myoepithelial cells whereas Cx26 is detected between secretory cells [15]. Similarly, these two connexins have also been studied in canines [16,17]. Studies of Cx43 expression comparing benign and malignant canine mammary tumors have shown a general decrease in expression in malignant tumors; however, histologically more aggressive or metastatic tumors may express higher Cx43 expression [15]. Therefore, upregulation of transcription of connexin genes or assembly of connexin proteins at the membrane has been proposed as a potential therapeutic target in humans. Hirschi et al. and Chen et al. proved that transfection of the Cx43 gene into cancer cells reverses the malignant phenotype of transformed cells to suppress human mammary carcinoma by decreasing growth rate and restoring the potential for cells to differentiate [7,18]. Therefore, this paper focuses on characterizing connexin proteins in both non-cancerous and cancerous cells of canine mammary carcinoma and examining the differential patterns of connexins in altering gap junctional intercellular communication.

## 2. Materials and Methods

### 2.1. Cell Lines

Wolfe et al. derived the CMT12 and CMT27 cell lines from canine mammary tumors [19]. CF41.Mg was derived from a mammary tumor of a 10-year-old, female beagle. CMEC was derived from a mammary gland tissue of an 8-year-old, female spayed Great Dane. Non-cancerous, canine mammary gland tissue was collected at the Kansas State Veterinary Health Center. All procedures were reviewed and approved by the Institutional Biosafety Committee. All cells were maintained in Dulbecco’s modified Eagle medium (DMEM, Life Technologies, Carlsbad, CA, USA) with 10% Fetal Bovine Serum (FBS, Atlas Diagnostics, Fort Collins, CO, USA).

### 2.2. Xenograft Tumor Model

Sixteen NU-Foxn1nu mice purchased from Jackson Labs weighing 14.5–20.1 g were randomly separated into 4 groups and injected in the inguinal mammary fat pad with 1 × 10^7^ CMT27, CMT12, or CF41.Mg cells or 1 × 10^6^ CMEC cells resuspended in 0.1 mL of serum free DMEM. Mice were monitored daily for tumor growth. Tumors were collected at time of euthanasia if the mass reached 1 cm in diameter; half of the tumor was fixed in 10% neutral buffered formalin for immunohistochemistry and the other half was homogenized with lysis buffer and used to quantify protein expression using western blotting.

### 2.3. Ethics Statement

Husbandry of animals was managed by the Comparative Medicine Group (CMG) at the College of Veterinary Medicine at Kansas State University in compliance with the U.S. National Research Council’s Guide for the Care and Use of Laboratory Animals. The CMG animal facilities are fully accredited by the Association for Assessment and Accreditation of Laboratory Animal Care, International (AAALAC). Animal care and use protocols were approved by the Institutional Animal Care and Use Committee (IACUC) at Kansas State University following NIH guidelines.

### 2.4. Immunofluorescence

CMT27, CMT12, CF41.Mg, and CMEC cells at 180,000 cells; 100,000 cells; 60,000 cells; and 60,000 cells, respectively, were seeded onto 35 mm glass bottomed microwell dishes and allowed to incubate overnight. The cells were fixed in 4% paraformaldehyde for 20 min, washed with phosphate buffered saline (PBS), incubated in 0.25% Triton X-100 in PBS for 20 min, and incubated in 3% bovine serum albumin (BSA) in PBS (blocking buffer) to block nonspecific binding. The cells were incubated overnight with primary antibody in BSA at a 1:500 dilution. The slides were then incubated with secondary antibody conjugated to Alexa Fluor 488 or 568 (Invitrogen, Eugene, OR, USA) at a 1:1000 dilution for an additional 2 h, and counter-stained with 1:1000 dilution Hoescht (Invitrogen, Eugene, OR, USA) in BSA. Images were captured using a Zeiss 880 confocal microscope. Primary antibodies were used against pan-cytokeratin (AE1/AE3, Santa Cruz Biotechnology, Dallas, TX, USA), calponin 1/2/3 (Santa Cruz Biotechnology, Dallas, TX, USA), Cx26 (Santa Cruz Biotechnology, Dallas, TX, USA), Cx32 (Zymed, San Francisco, CA, USA), Cx36 (Zymed, San Francisco, CA, USA), Cx43 (Santa Cruz Biotechnology, Dallas, TX, USA), Cx45 (Chemicon International, Temecula, CA, USA), and Cx46 (Santa Cruz Biotechnology, Dallas, TX, USA). All primary antibodies were used at 1:500 dilution. CMEC cells were tested for epithelial markers, pan-cytokeratin for luminal epithelial cells and calponin for myoepithelial cells.

### 2.5. Western Blot Analysis

Cells grown in T-75 cm^2^ culture flasks at 90% confluence were harvested with lysis buffer with a 1:1000 dilution of protease inhibitor (Sigma-Aldrich, Saint Louis, MO, USA) and centrifuged at 13,200 rpm for 30 min at 4 °C. Forty micrograms of whole cell extract (WCE), from cell culture and from the xenograft tumors, was resolved in 10% TGX precast gels using the SDS-polyacrylamide gel electrophoresis (PAGE) method. The gels ran at 110 V for 65 min in Tris/Glycine/SDS (TGS) buffer and were then transferred to polyvinylidene difluoride (PVDF) membranes (Bio-Rad, Hercules, CA, USA) using the Bio-Rad Trans-Blot Turbo. The membranes were blocked with 1% milk/TBST (Tris-buffered saline with 0.1% Tween 20) for 40 min and incubated overnight with a 1:500 dilution of primary antibody in 1% milk/TBST. The following day, the membranes were incubated with secondary antibody conjugated to horseradish peroxidase (Cell Signaling, Technology, Beverly, MA, USA) at a 1:1000 dilution for two hours and then developed using ChemiGlow West Chemiluminescence Substrate Kit (ProteinSimple, San Jose, CA, USA) and visualized using the FluorChemE (ProteinSimple, San Jose, CA, USA) imaging system. Beta tubulin was used as a loading control. Data was analyzed using AlphaView Software 3.2 (ProteinSimple, San Jose, CA, USA). Significance was considered at a *p*-value ≤ 0.05 using Student’s *t*-test analysis. The same antibodies previously mentioned were used here along with beta tubulin (Santa Cruz Biotechnology, Dallas, TX, USA).

### 2.6. Scrape Load/Dye Transfer Assay

Cells were seeded as a 100% confluent monolayer onto glass coverslips. 1% Rhodamine dextran and 1% Lucifer yellow dyes were mixed at a 1:1 ratio and 2 µL of the mixture was placed onto coverslip. A p10 pipette tip was used to scrape the coverslip. After 3 min of incubation with dye mixture, the slides were washed with PBS and incubated with 10% DMEM for 20 min. The slides were again washed with PBS three times and incubated with 4% paraformaldehyde for 30 min and then mounted onto slides for imaging.

### 2.7. Immunohistochemistry

Twelve canine mammary tumor biopsies, characterized as solid carcinomas, and five canine mammary gland tissues, embedded in paraffin were obtained from archives of the Kansas State University Veterinary Diagnostic Laboratory within 2017 and 2019. The breeds included an English setter, German shepherd, rat terrier, miniature pinscher, Great Dane, German shorthair pointer, schnauzer, Labrador retriever, shih tzu, beagle, mixed breed, and one unknown. The xenograft tumors collected from the mice were fixed in 10% neutral buffered formalin and also embedded in paraffin. Each block was cut into 5 µm sections and mounted onto positively charged slides. After heating the slides at 65 °C for 25 min, the sections were deparaffinized in three xylene washes for five minutes each and rehydrated in two 100% ethanol and two 90% ethanol baths for 15 min each. A steaming citrate bath was used for antigen unmasking for 20 min, cooled for another 20 min, and incubated for 5 min in 1% hydrogen peroxide. The histological sections were blocked for 1 h using 5% horse serum in PBS and incubated overnight with primary antibody at a 1:50 dilution in 5% horse serum. The sections were then incubated with biotin-conjugated secondary antibody at a 1:100 dilution in 1.5% blocking serum for one hour and washed three times in PBS. The histological sections were then incubated for 30 min with the Vectastain Elite ABC kit (Vector Laboratories, Burlingame, CA, USA), incubated with DAB (Vector Laboratories, Burlingame, CA, USA) for 20 min, and counterstained in Gill’s Hemaxotylin Solution (Vector Laboratories, Burlingame, CA, USA) for 4 min, and mounted with permount (Fisher Scientific, Fair Lawn, NJ, USA). Primary antibodies against connexin 43 (Santa Cruz Biotechnology, Dallas, TX, USA) and connexin 26 (Santa Cruz Biotechnology, Dallas, TX, USA) were used in this case. Secondary antibodies were used against rabbit and mouse IgG (Vector Laboratories Inc., Burlingame, CA, USA).

## 3. Results

### 3.1. Connexin Expression in CMEC and CMT Cells

All mammary cells were grown in 10% DMEM conditioning media at 60% confluency. Cells were fixed and immunofluorescence assay was performed using connexin antibodies. Connexins 26, 32, 36, 43, 45, and 46 were selected for the study and Table 1 lists the tissues they are expressed in humans and rodents [5,20,21] The results show that connexins 26 and 43 were present in all cells but were mostly localized in the nucleus, especially Cx43 (Figure 1A). All cells expressed undetectable levels of connexins 45, 46, 36, and 32, except for weak Cx36 positivity in CF41.Mg (Figure 1B,C). Three independent experiments of triplicate western blot measurements revealed a significant decrease of Cx26, Cx32, and Cx45 in CMT12, and Cx36 and Cx43 in both CMT12 and CMT27 (Figure 2).

To determine whether the connexins present in the cell lines were capable of forming functional gap junctions, the scrape load/dye transfer assay was performed. Lucifer yellow dye, with a molecular weight of 457 Daltons, has the ability to pass through gap junctions; therefore, the more cells that the dye passes through correlates with the number of functional gap junctions present. The assay showed that Lucifer yellow passes through three adjacent CMEC cells bidirectionally from the scrape, whereas CMT12 and CMT27 cells passed dye through a single cell and CF41.Mg passed dye through two to three cells, in the same time frame (Figure 3). These results suggest that cancerous cells, CMT12 and CMT27, have relatively lower gap junction capacity than CMEC non-cancerous cells.

### 3.2. Connexin Expression in Canine Mammary Cells Xenotransplanted to Nude Mice

To assess the expression of connexins in vivo, ten million cells were injected subcutaneously into the right inguinal region of the mammary fat pad. CMECs were also implanted not only to serve as a control but also to ensure that non-cancerous cells, CMECs, have not been transformed. No tumor development was detected in CMEC mice. All CMT27 mice developed tumors that measured one centimeter in diameter within 68 days (Figure 4A). Histological evaluation of the CMT27 xenograft tumors showed a solid nest of predominately poorly-differentiated tumor cells, exhibiting high cellularity with small, pleomorphic, epithelial cells with prominent nucleoli, coarse granular chromatin, and high mitotic activity (Figure 4C). Neoplastic cells were larger than normal epithelium with a characteristic epithelioid morphology. With immunohistochemistry these tumors stained positive for Cx26 (Figure 4D) but there were undetectable levels of Cx43 (Figure 4E). Control samples of other tissues were used to validate Cx26 and Cx43 antibodies. Canine tumor tissue was stained negative for Cx43 (Figure 4F), mouse liver tissue was stained positive for Cx26 (Figure 4G), and mouse heart tissue was stained positive for Cx43 (Figure 4H). Immunoblotting of the tumor lysates showed variable Cx26 expression which is consistent with the cell-based data (Figure 4B). The variation in connexin expression for Figure 2, Figure 3 and Figure 4 is observed. This in part is due to the sensitivity of the antibodies used in three different techniques. Three CMT12 mice had no visible tumor development and were euthanized after 57 days; the other CMT12 mouse developed a small tumor about 0.4 cm in diameter and was euthanized after 77 days. Two CF41.Mg mice developed tumors about 0.2–0.3 cm in diameter and the other two had no tumor development after 57 days. These results suggest that implantation of CMT27 into nude mice can be used as a xenograft model to study canine mammary cancer.

### 3.3. Immunohistochemistry of Canine Mammary Tumors

To evaluate connexin expression in dogs with mammary carcinoma, immunohistochemistry (IHC) was performed in 12 canine mammary carcinoma biopsies. Two of these were intact females and the other ten were spayed females. While Cx26 showed positive immunostaining in all samples, only 25% of samples showed Cx43 positivity; representative images from two dogs are shown (Figure 5). Hematoxylin and esosin staining was performed for all samples (Figure 5C). Cx26 expression was seen in both epithelial and mesenchymal cells varying from a membranous (Figure 5B) to strong cytoplasmic (Figure 5E) expression in tumor tissues. Cx43 positivity was confined to the cytoplasm of epithelial cells (Figure 5F). Normal mammary gland tissue was also obtained and stained positive for Cx26 and Cx43 (Figure 5G,H). Interestingly, both samples from the intact females were Cx43 positive which may suggest a phenotypic pattern for connexin expression in intact females compared to spayed females (Table 2). These results suggest that although Cx26 may be widely expressed in mammary tumors, Cx43 is expressed in only a small subset of these tumors.

## 4. Discussion

Mammary neoplasia is the most common diagnosed tumor in bitches and thus they present a significant clinical challenge. Many biomarkers of mammary neoplasia have been reported to better understand the disease in part for early detection and prognosis [21]. In literature, neoplastic cells have a loss of intercellular communication, particularly gap junction proteins (connexins) [4]. Here we provide for the first time that the differential pattern of connexins are presented in both non-cancerous and cancerous cells of canine mammary carcinoma. Therefore, studying gap junctions and their role in mammary cancer may provide insight into cell communication and possibly a novel therapeutic approach. Gap junctions are vital to intercellular communication in that they allow homeostatic processes such as the cell cycle to function normally so that cells may grow in a coordinated manner and at an appropriate rate. Currently, there is little knowledge of cell communication in canine mammary carcinoma beyond histological data. Due to the nonselective nature of gap junctions, connexin proteins provide a unique target for therapy; if normal gap junction function can be restored, cancer cells may be able to communicate with surrounding normal cells and subsequently alter its microenvironment.

The use of non-cancerous cells from the mammary gland of a dog should provide valuable insight into the changes of cell communication, particularly the loss of gap junction activity during cancer initiation, compared with the established cell lines from canine mammary carcinoma. The benefits of having non-cancerous canine mammary cells would provide a model to examine specific cellular targets under non-disease stage. Previously, non-cancerous cells were identified with calponin and pan-cytokeratin epithelial markers (Appendix A). Without immortalizing the cells, CMEC could only be studied for seven passages and therefore, the number of cells available for studies was limited. However, it is critical to include CMEC as a baseline for the study in comparison of gap junction protein expression and function against the cancerous mammary cells.

Immunofluorescence revealed that all cells are positive for both connexins 26 and 43 but the connexins appear to be more localized in the nuclei instead of the membrane. It is possible that canine mammary carcinoma cells may have defects in transportation of the connexins to the membrane where gap junctions form rather than defects in transcription and translation [22,23]. Furthermore, GJIC defects can be examined by the loss of dye movement across gap junction channels. Subsequent studies using the scrape load/dye transfer assay showed less dye transfer between adjacent cells in CMT12 and CMT27 compared to CMEC, suggesting that gap junction capacity is higher in CMEC. CF41.Mg cells showed similar dye transfer compared with CMEC and also showed similar connexin protein expression levels and may suggest that these cancerous cells may not have a loss of GJIC capacity similar to CMT12 and CMT27 cells. Western blot results showed decreased Cx26 in CMT12 and decreased Cx43 in both CMT12 and CMT27 compared to CMEC which is further supported by RT-PCR results from Gotoh et al. and immunostaining results from Torres et al. [16,17]. There was also a significant decrease in Cx32, 36, and 45 in CMT12 and Cx36 in CMT27; these connexins have not been well-studied in regards to the mammary gland and more research needs to be done to conclude whether these results may be of importance.

All CMT27 mice developed tumors, but only one of four CMT12 mice developed a small tumor. While CF41.Mg has been established as a metastatic canine mammary tumor cell line [24,25], only two of the mice developed small tumors. Therefore, the more mesenchymal CF41.Mg may not be an appropriate cell line for studies involving primary tumors but may be better for studying epithelial-to-mesenchymal transition and metastasis. CMT27-xenograft tumors revealed positive staining for Cx26 and undetectable Cx43. Western blot was positive for both connexins; in our cell-based experiments, CMT27 expressed significantly lower levels of Cx43 compared to CMEC. However, results of immunochemistry sections revealed positive for Cx26 and negative for Cx43, in part due to the sensitivity of antibodies for immunochemistry assay.

Overall, this paper is the first to compare connexin protein expression levels in non-cancerous and cancerous mammary cells derived from dogs. Future studies may include whether the change of Cx43 level in CMT12 and CMT27 would alter the neoplastic phenotype, as it has been shown in human mammary carcinoma [7,18]. Furthermore, the mode of action in the loss of Cx43 may occur at either the transcriptional level or during protein transport and assembly [22,23]. Further studies into the exact mechanism of gap junction inactivation are required to pinpoint the particular cellular pathway affected in canine mammary carcinoma cells; studies may possibly include measuring connexin recycling time and ubiquitinylation, and level of mRNA expression.

## 5. Conclusions

CMT12 and CMT27, canine mammary tumor cells, have lower gap junction activity compared to CMEC non-cancerous cells. There is a distinct differential pattern of connexin 26 and connexin 43 in non-cancerous mammary cells, CMEC, and cancerous mammary cells, CMT12, CMT27, and CF41.Mg. All CMT27-xenotransplanted mice developed tumors and each tumor was positive for connexin 26. Mammary carcinoma from 12 dogs showed that connexin 26 was expressed in both epithelial and myoepithelial cells varying from membranous to strong cytoplasmic.

## Figures and Tables

**Figure 1 vetsci-06-00101-f001:**
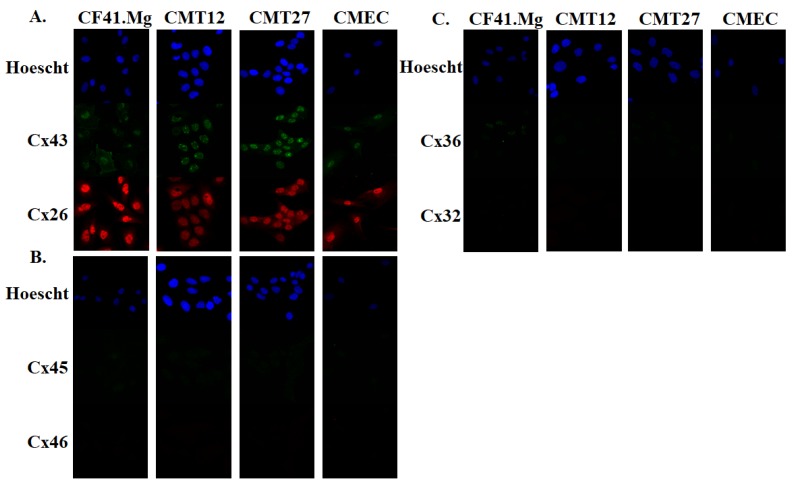
Immunofluorescence of connexin proteins. Immunofluorescence images of connexins 43 and 26 (**A**), connexins 45 and 46 (**B**), and connexins 36 and 32 (**C**). Hoescht stain was used to stain nuclei. All images were taken at 40X on a Zeiss 880 confocal microscope.

**Figure 2 vetsci-06-00101-f002:**
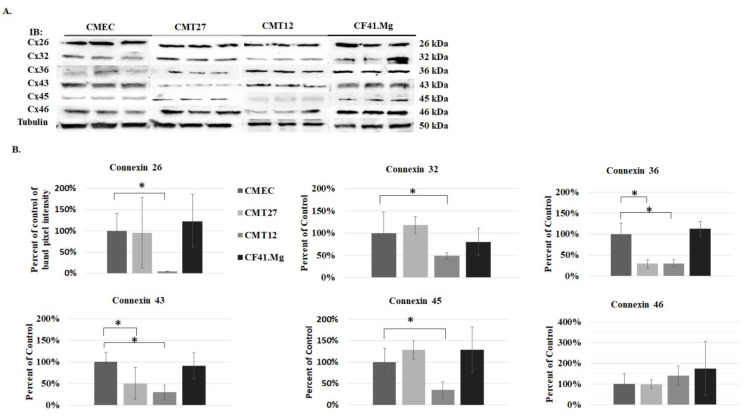
Examination of connexin protein expression. Representative western blot images of connexin expression (**A**) and quantitative analysis (**B**). Statistical analysis was performed, indicating * at a *p*-value ≤ 0.05 using Student’s *t*-test analysis and error bars represent standard error of the mean of three independent experiments.

**Figure 3 vetsci-06-00101-f003:**
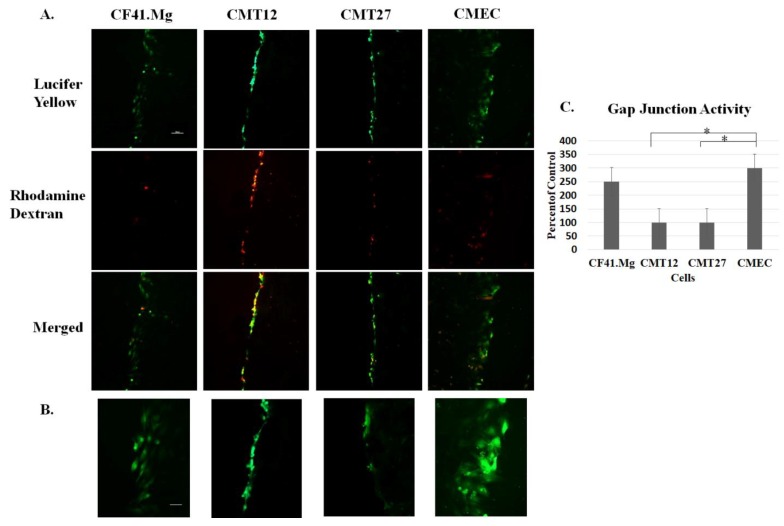
Examination of gap junction activity. The scrape load/dye transfer assay shows passage of Lucifer yellow dye through cells with functional gap junctions, the scrape indicated by rhodamine dextran dye as a negative control, and a merged image, all taken at 10X. The bar is equivalent to 100 μm. (**A**). A closer view at 20X of the Lucifer yellow inside the cells. The bar is equivalent to 20 μm (**B**). Graphical representation of the levels of gap junction activity comparing the four cells (**C**). Statistical analysis was performed, indicating * at a *p*-value ≤ 0.05 using Student’s *t*-test analysis and error bars represent standard error of the mean of three independent experiments.

**Figure 4 vetsci-06-00101-f004:**
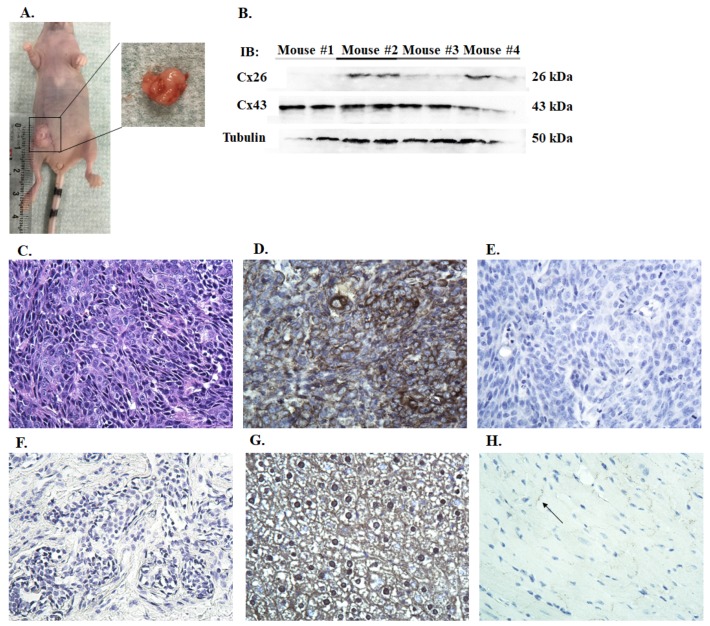
Evaluation of connexin expression in xenograft tumors. A female, nude mouse injected with CMT27 cells into the inguinal mammary fat pad developed a xenograft tumor about 1 cm in diameter (**A**). Western blot of the four CMT27 tumors for Cx26 and Cx43 expression (**B**). Representative images of one of the tumors stained with hematoxylin and eosin (**C**), Cx26 (**D**), and Cx43 (**E**). Negative control using canine tumor tissue (**F**) and positive controls for Cx26 using mouse liver (**G**) and for Cx43 using mouse heart (**H**) are shown at 40X. Arrow indicates the positive staining of Cx43 in myocardium.

**Figure 5 vetsci-06-00101-f005:**
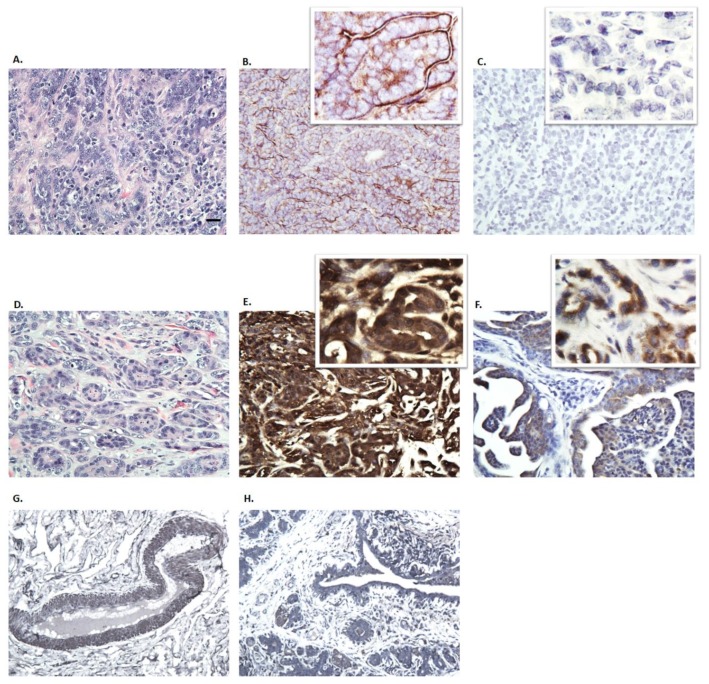
Representative images of immunohistochemistry from two dogs with tumors and a dog with no tumor, stained with hematoxylin and eosin (**A**,**D**) and immunostaining for Cx26 (**B**,**E**,**G**), and Cx43 (**C**,**F**,**H**). Samples from dogs with tumors, Cx26 was expressed in both epithelial and myoepithelial cells (**B**,**E**) but expression varied from membranous (**B**) to strong cytoplasmic (**E**). Normal mammary gland tissue was stained for Cx26 and Cx43 (**G**,**H**). Cx43 had cytoplasmic expression confined to the epithelial cells. Panel **A–F** images were taken at 40X with small window image at 60X; and panels G and H images were taken at 20X.

**Table 1 vetsci-06-00101-t001:** Summary of organ/tissue distribution of the connexins studied.

Connexin	Organ/Tissue Distribution
Cx26	Liver, mammary gland, pancreas, skin, lung, brain, and endometrium
Cx32	Liver, brain, uterus, thyroid, kidney, and pancreas
Cx36	Neuronal cells of the central nervous system
Cx43	Epithelia, heart, lens, bone, skin, pancreas, uterus, gonads, bone, connective tissue, and brain
Cx45	Heart, smooth muscle, and neurons
Cx46	Heart, lens, kidney, lung, bone, and testes

**Table 2 vetsci-06-00101-t002:** Summary of immunohistochemistry of spontaneous mammary carcinoma biopsies from 12 dogs including breed, spayed, or intact status; and immunoreactivity of Cx26 and Cx43.

Summary of Immunohistochemistry				
Dog	Breed	Spayed/Intact	Cx26	Cx43
1	English setter	Spayed	+	−
2	German shepherd	Spayed	+	−
3	Unknown	Intact	+	+
4	Rat terrier	Spayed	+	−
5	Miniature pinscher	Spayed	+	−
6	Great Dane	Spayed	+	−
7	German shorthaired pointer	Spayed	+	−
8	Mixed breed	Intact	+	Weak +
9	Schnauzer	Spayed	+	+
10	Labrador retriever	Spayed	+	−
11	Shih Tzu	Spayed	+	−
12	Beagle	Spayed	Weak +	−
Overall positivity			100%	25%

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
