# Peer review of "Connexin 26 and Connexin 43 in Canine Mammary Carcinoma"

_vetsci, 2019, doi:10.3390/vetsci6040101_

Round 1

Reviewer 1 Report

Overall very interesting paper describing the research conducted to determine the differential expression of connexins between normal and mammary cancer cells in dogs. This study will contribute towards a better understanding of the progression of mammary carcinoma in dogs 

I suggest some minor edits:-

The references need to be in superscript throughout the paper. Line 1  : Is there additional epidemiological evidence of the high incidence of mammary cancer in dogs besides in Italy? How about in the US or in other European countries? Introduction, second paragraph, connexins are repeatedly misspelt as "connexons" Materials and Methods, 2.2, last sentence should end with "Western blotting", not "Western blot". Materials and Methods, 2.3, part of the ethics statement is in a different font from the rest of the paper 2.5 Western Blot Analysis "cm2" - 2 should be subscript 2.7 Immunohistochemistry - penultimate sentence should end with "were used in this case" rather than "were used here". Results, 3.1, paragraph 1 second sentence "cells were fixed and an immunofluorescence assay was performed using connexin antibodies." Results, 3.1, paragraph 2 last sentence "These results suggest that the cancerous cell lines CMT12 and CMT27 have a relatively lower gap junction capacity than CMEC non-cancerous cells." Discussion, paragraph 3, sentence 2. This sentence is highly ambiguous and needs to be rewritten more clearly - what do the authors mean by "allow a foundation to be built" Discussion, paragraph 4, sentence 3. "GJIC defects" rather than "the defect of GJIC". Discussion, paragraph 5, sentence 2. Delete 'however' and start the sentence with "While CF41.Mg has been.....". Discussion, last paragraph, last sentence. This sentence has no place in the conclusion to the paper and should be placed in the introduction, perhaps in the second paragraph.

Author Response

The references need to be in superscript throughout the paper.

Response: All the reference numbers have been changed to superscript throughout the paper.

Line 1: Is there additional epidemiological evidence of the high incidence of mammary cancer in dogs besides in Italy? How about in the US or in other European countries?

Response: Thank you for these questions.  I have included additional evidence for the US and Mexico. A following statement has added. “A retrospective study, focused on 11,544 dog biopsy from January 2002 to December 2012, demonstrated that mammary tumors in female dogs are a major health problem and approximately 50% are malignant.”

Introduction, second paragraph, connexins are repeatedly misspelt as "connexons"

Response: Thanks for pointing out the spelling of connexins and connexons.  Interestingly, six connexins make up one connexon.  Therefore, the word “connexon” was used to indicate that the hemichannel, connexon, is present and not a single protein, connexin.  In other words, the usage of connexon is to indicate that multiple connexins are present to form the hemichannel, called connexon.

Materials and Methods,2.2, last sentence should end with "Western blotting", not "Western blot".

Response: Thank you.  A correction has been made.

Materials and Methods, 2.3, part of the ethics statement is in a different font from the rest of the paper

Response:  Thank you.  A change has been made.

5 Western Blot Analysis "cm2" - 2 should be subscript

Response: Agree. A correction has been made.

7 Immunohistochemistry - penultimate sentence should end with "were used in this case" rather than "were used here".

Response: Thanks. A change has been made.

Results, 3.1, paragraph 1 second sentence "cells were fixed and an immunofluorescence assay was performed using connexin antibodies."

Response: Thank you. 

Results, 3.1, paragraph 2 last sentence "These results suggest that the cancerous cell lines CMT12 and CMT27 have a relatively lower gap junction capacity than CMEC non-cancerous cells."

Response: Thank you.  A correction has been made.

Discussion, paragraph 3, sentence 2. This sentence is highly ambiguous and needs to be rewritten more clearly - what do the authors mean by "allow a foundation to be built"

Response: The authors are excited to obtain primary cells of non-cancerous mammary cells. The sentence has been changed to “The benefits of having non-cancerous canine mammary cells would provide a model to examine specific cellular targets under non-disease stage.”

Discussion, paragraph 4, sentence 3. "GJIC defects" rather than "the defect of GJIC".

Response: Thanks. A change has been made.

Discussion, paragraph 5, sentence 2. Delete 'however' and start the sentence with "While CF41.Mg has been.....".

Response: Thank you. Corrections have been made.

Discussion, last paragraph, last sentence. This sentence has no place in the conclusion to the paper and should be placed in the introduction, perhaps in the second paragraph. 

Response:  Agree. This sentence has been deleted.

Reviewer 2 Report

In general, the paper is well written. The only suggestion I have is to examine additional features of the clinical canine mammary tumors examined to determine what is correlated with connexin status. For instance, what were the mitotic rates of the tumors? Were any invasive? Did any have metastasis to regional lymph nodes?

Author Response

In general, the paper is well written. The only suggestion I have is to examine additional features of the clinical canine mammary tumors examined to determine what is correlated with connexin status. For instance, what were the mitotic rates of the tumors? Were any invasive? Did any have metastasis to regional lymph nodes?

Response: Excellent question.  Interestingly, we found that vimentin (biomarker for epithelial-to-mesenchymal transition) was highly expressed in all these tumors, indicating that all 12 mammary carcinoma biopsies are invasive.

Reviewer 3 Report

Veterinary Sciences 572759

Connexin 26 and connexin 43 in canine mammary carcinoma

Introduction:

The information regarding the incidence of canine mammary tumors (CMT) seems to me to be excessive and of little relevance to the purpose of this study and, therefore, should be reduced.

1st paragraph, line 1 – “…neoplasia in female dogs…” instead of “…cancer in female dogs…”

2nd paragraph, lines 2-5 – references missing

2nd paragraph, line 8 – “neoplastic initiation” instead of “cancer formation”

3rd paragraph, lines 2-3 – “Different connexin…tissue-specific”. Rephrase please

3rd paragraph, line 10 – “histologically more aggressive” instead of “more histologically aggressive” and rephrase “…may express higher cx43 expression”

Regarding the aims of the study, the authors must emphasize the features that distinguish the present investigation from others previously performed on the same issue.

Material and Methods:

2.1. Origin and references on cell lines CF41.Mg and CMEC are missing.

2.2. 1x107 and 1x106 instead of 1x107 and 1x106, respectively.

2.2. The authors do not provide explanations for rejecting xenograft tumors < 1cm.

2.3. Attention to font size (lines 3-4).

2.4. What is the rationale beneath the use of markers for pancytokeratins, calponin, cx32, cx36, cx45 and cx46? This investigation shown little consistency at this point, as these markers are not used in xenografts or spontaneous canine mammary tumors. Besides this data is not focused on the discussion section.

2.7. The authors do not explain the criteria used to select spontaneous tumor cases. It is essential to characterize and classify properly the spontaneous CMT according to the histological classification systems commonly used by veterinary pathologists (Misdorp et al 1999, Goldschmidt et al 2011). Are they benign or malignant lesions? Simple/complex/mixed tumors?

Results:

The content of tables 1 and 2 is reversed.

3.2. Full characterization of the xenograft tumors is needed (benign/malignant tumors; composed solely of luminal epithelium or including neoplastic myoepithelium?)

3.2. Necrotic areas must be avoided in immunohistochemical examination because of non-specific staining.

Figs 4 (F, G and H) should be deleted.

Fig 4K is not a good representative image of a positive control. I recommend that it should be replaced by another, more enlightening image, where the immunostaining is obvious.

3.3. Lines 1-5 correspond to materials and methods, not to results.

3.3. The authors do not describe the expression pattern for cx26 and cx43 (membranar, cytoplasmic, nuclear; immunostaining extension; luminal epithelium and/or myoepithelium staining; immunostaining distribution).

3.3. (lines 7-9) – “Interestingly…to spayed females”. It is not possible to reach this conclusion with only 2 cases.

3.3. Figure 5F is of poor quality and does not clearly depict the immunostaining distribution and intracelular location.

Veterinary Sciences is a journal mainly devoted to veterinary medicine with a main focus on animals. In this sense, the considerations presented by the authors regarding the importance of CMT as suitable models for the study of human breast cancer seem to me excessive and should thus be reduced. Besides, the authors use very old references (all, except one, are from 25-30 years ago) to justify the relevance of CMT as a model for breast cancer. Independently of their suitability as models for breast cancer, it is important to evaluate cell adhesion/communication mechanisms in CMT in order to explore novel therapeutic approaches in this animal species.

Discussion:

2nd paragraph, lines 3-4: the authors state that “Currently, there is little knowledge of cell communication in canine mammary carcinomas”. I do not totally agree with the authors. A simple search at Pubmed allows you to find dozens of papers on this topic, published in the last 3 decades.

2nd paragraph, lines 5-6: “…if normal gap junction function can be restored…return to a normal phenotype”. This statement seems to me very naïve and is not supported by recent credible evidence.

3rd paragraph, lines 2-3: “cancer initiation” rather than “cancer formation”

Differences observed in the expression of markers cx26 and cx43 may simply be a reflex of the characteristics of the cell lines included in the study or of the histotype of the spontaneous neoplasms (in this sense it is essential to clarify the histogenesis of the neoplastic cells and to determine their origin - luminal epithelial or myoepithelial). As the authors refer in the Introduction section (3rd paragraph, lines 6-7)cx 26 is expressed in luminal epithelial cells, while cx43 is expressed in myoepithelial ones.

It would also be interesting to evaluate the expression of these markers in normal canine mammary tissue samples (not just normal cell lines) to compare with the results observed in spontaneous mammary tumors.

References:

Attention must be given to the references throughout the text (it is necessary to standardize its presentation - superscript).

The references seem very old to me (15 references are over 20 years old and only 11 are from this decade) which is not understandable since this topic has been the subject of several research in recent years.

Author Response

The information regarding the incidence of canine mammary tumors (CMT) seems to me to be excessive and of little relevance to the purpose of this study and, therefore, should be reduced.

Response: Thanks for the suggestion. A significant reduction in the introduction has been made.

1stparagraph, line 1 – “…neoplasia in female dogs…” instead of “…cancer in female dogs…”

Response: A change to “neoplasia” has been made.

2ndparagraph, lines 2-5 – references missing

Response: A reference has been added to this statement.

2ndparagraph, line 8 – “neoplastic initiation” instead of “cancer formation”

Response: A change has been made.

3rdparagraph, lines 2-3 – “Different connexin…tissue-specific”. Rephrase please

Response: Thank you. The statement has been changed to “Different types of connexin are expressed as cells undergo differentiation; furthermore, the patterns of expression are cell- and tissue-specific.”

3rdparagraph, line 10 – “histologically more aggressive” instead of “more histologically aggressive” and rephrase “…may express higher cx43 expression”

Response: Thanks.  A change has been made.

Regarding the aims of the study, the authors must emphasize the features that distinguish the present investigation from others previously performed on the same issue.

Response: Agree; therefore, a statement summarizes the significance of the paper in which for the first time the differential pattern of connexin proteins are shown in both non-cancerous and cancerous cells of canine mammary carcinoma.

1. Origin and references on cell lines CF41.Mg and CMEC are missing.

Response: Thank you. The information have been added. CF41.Mg was purchased from American Type Culture Collection and CMEC was established at Kansas State University, College of Veterinary Medicine. These cells were isolated and screened for characteristics of epithelial cells using standard biomarkers for epithelial functions, including calponin and pan-cytokeratin.

2. 1x107and 1x10instead of 1x107 and 1x106, respectively.

Response: These grammatical errors have been corrected.

2. The authors do not provide explanations for rejecting xenograft tumors < 1cm.

Response: Thanks for pointing out this procedure. The Comparative Medicine Group of the Institutional Research Compliance Office recommends to minimize pain due to tumor burden; thus, tumor size of 1 cm in length was set to be the point of euthanasia. 

3. Attention to font size (lines 3-4).

Response: Thank you.  A change has been made.

4. What is the rationale beneath the use of markers for pancytokeratins, calponin, cx32, cx36, cx45 and cx46? This investigation shown little consistency at this point, as these markers are not used in xenografts or spontaneous canine mammary tumors. Besides this data is not focused on the discussion section.

Response: Thank you for this question.  We failed to mention that CMECs were isolated at Kansas State University and characterized by screening cell population with biomarker calponin and pan-cytokeratin. We have added a statement in the Materials and Methods 2.1 to reflect this rationale for using pan-cytokeratins and calponin as makers. These makers were not used in xenograft tumor experiment because these cells have already screened for these markers prior to the implantation.

7. The authors do not explain the criteria used to select spontaneous tumor cases. It is essential to characterize and classify properly the spontaneous CMT according to the histological classification systems commonly used by veterinary pathologists (Misdorp et al 1999, Goldschmidt et al 2011). Are they benign or malignant lesions? Simple/complex/mixed tumors?

 Response: Thank you for the suggestion. We have added the criteria in selecting the spontaneous tumor cases in Section 2.7. The criteria in selecting 12 mammary tumor biopsies from 2017 archive were malignant tumors of solid carcinomas.

The content of tables 1 and 2 is reversed.

Response: Thank you. Correction has been made.

2. Full characterization of the xenograft tumors is needed (benign/malignant tumors; composed solely of luminal epithelium or including neoplastic myoepithelium?)

Response: Full characterization of the xenograft tumors has been added in Section 3.2. Tumor sections showed a sold nest of predominately poorly-differentiated tumor cells with large, irregular nuclei, coarse granular chromatin, prominent nucleoli, and high mitotic activity. Neoplastic cells were larger than normal epithelium with a characteristic epithelioid morphology and marked nuclear pleomorphism.

2. Necrotic areas must be avoided in immunohistochemical examination because of non-specific staining.

Response: Agree; thus, we noted that the positive stain for Cx43 at necrotic areas of the tumors.   

Figs 4 (F, G and H) should be deleted.

Response: We think it’s important to provide this observation to the readers.

Fig 4K is not a good representative image of a positive control. I recommend that it should be replaced by another, more enlightening image, where the immunostaining is obvious.

Response: The pattern and expression of Cx43 of the heart tissue has been used as the gold standard for a positive control in immunostaining of Cx43. Thus, the intensity of the staining is relevant to the study at the point of the experiment.

3. Lines 1-5 correspond to materials and methods, not to results.

Response: The information has been moved to the Materials and Methods section.

3. The authors do not describe the expression pattern for cx26 and cx43 (membranar, cytoplasmic, nuclear; immunostaining extension; luminal epithelium and/or myoepithelium staining; immunostaining distribution).

Response: A description of connexin expression in immunostaining distribution has been added in Section 3.3. “Cx26 expression was seen in both epithelial and mesenchymal cells varying from a membranous (Figure 5B) to strong cytoplasmic (Figure 5E) expression.” Additional information was added to Figure 5. Dr. Sarah Schneider, anatomical pathologist, has been added to the authorship for her contribution to this manuscript.

3. (lines 7-9) – “Interestingly…to spayed females”. It is not possible to reach this conclusion with only 2 cases.

Response: We understand your point referring to sample size of 2; however, the overall sample size of 12 biopsies provided an interesting phenotypic pattern for connexin expression. This conclusion is based on the sample size of 12 biopsies.

3. Figure 5F is of poor quality and does not clearly depict the immunostaining distribution and intracelular location.

 Response: 60X images have been added to this figure. The legend was modified to reflect these changes.

Veterinary Sciences is a journal mainly devoted to veterinary medicine with a main focus on animals. In this sense, the considerations presented by the authors regarding the importance of CMT as suitable models for the study of human breast cancer seem to me excessive and should thus be reduced. Besides, the authors use very old references (all, except one, are from 25-30 years ago) to justify the relevance of CMT as a model for breast cancer. Independently of their suitability as models for breast cancer, it is important to evaluate cell adhesion/communication mechanisms in CMT in order to explore novel therapeutic approaches in this animal species.

Response: Thank you for this suggestion. We have deleted these sentences. The focus has been changed and references have been updated.

Mammary neoplasia is the most common diagnosed tumor in bitches and thus they present a significant clinical challenge. Many biomarkers of mammary neoplasia have been reported to better understand the disease in part for early detection and prognosis. Here we provide for the first time that the differential pattern of connexin proteins are presented in both non-cancerous and cancerous cells of canine mammary carcinoma. Therefore, studying gap junctions and their role in mammary cancer may provide insight into cell communication and possibly a novel therapeutic approach.

2ndparagraph, lines 3-4: the authors state that “Currently, there is little knowledge of cell communication in canine mammary carcinomas”. I do not totally agree with the authors. A simple search at Pubmed allows you to find dozens of papers on this topic, published in the last 3 decades.

Response: Correction has been made to reflect the need to examine cell communication at cellular level beyond histological examination of canine mammary carcinomas.

2ndparagraph, lines 5-6: “…if normal gap junction function can be restored…return to a normal phenotype”. This statement seems to me very naïve and is not supported by recent credible evidence.

Response: Thank you for pointing out this naïve statement. The complexity of cancer formation is beyond one pathway alteration. Thus, a change has been made to reflect the complexity of cancer and its microenvironment.

3rdparagraph, lines 2-3: “cancer initiation” rather than “cancer formation”

Response: A change has been made.

Differences observed in the expression of markers cx26 and cx43 may simply be a reflex of the characteristics of the cell lines included in the study or of the histotype of the spontaneous neoplasms (in this sense it is essential to clarify the histogenesis of the neoplastic cells and to determine their origin - luminal epithelial or myoepithelial). As the authors refer in the Introduction section (3rdparagraph, lines 6-7)cx 26 is expressed in luminal epithelial cells, while cx43 is expressed in myoepithelial ones.

Response: Thank you for this insight. We have added the description of cx26and cx43 to reflect this suggestion.

It would also be interesting to evaluate the expression of these markers in normal canine mammary tissue samples (not just normal cell lines) to compare with the results observed in spontaneous mammary tumors. Attention must be given to the references throughout the text (it is necessary to standardize its presentation - superscript).

Response: All the changes have been made.

The references seem very old to me (15 references are over 20 years old and only 11 are from this decade) which is not understandable since this topic has been the subject of several research in recent years.

Response: Some references have been updated to reflect more recent studies.

Round 2

Reviewer 3 Report

I would like to thank the authors for their efforts and commitment in trying to answer all the questions raised in the revision of the paper, having corrected most of the minor concerns. There are, however, some minor points that still deserve attention:

- based on the way the authors describe the methods (2.2. Xenograft tumor model) I got the (wrong) idea that only tumors> 1cm were included in the study, and those smaller than 1cm were excluded. I suggest you to rephrase this part so that no doubt remains.

- the authors refer they included a statment explaining the rationale for using pan-cytokeratins and calponin markers in the M&M section (2.1. Cell lines) but I cannot find such statement.

- the first paragraph of the discussion is not well written; some logic is missing. Rephrase please.

- 2nd paragraph (last lines) - the statement is poor and lacks bibliographic support. Rephrase please.

Nonetheless, the major weakness of this paper lies in the histopathological and immunohistochemical study. The absence of mention to the immunohistochemical study of xenografts and spontaneous canine mammary tumors in the “Aims of the study” reinforces my conviction that this study was not delineated with this component in mind.

I am not convinced of the reliability of immunohistochemical labeling with the Cx43 antibody. The positive control used for Cx43 shows no obvious staining (Fig 4K) and therefore cannot validate the experiments performed with this marker. Take, for example, photos 4E, 4H and 4K (all related to Cx43 marking). The authors state that in Fig 4H there is no immunostaining (I totally agree) but in Figs 4E and 4K they refer to positive labelling. However, the "immunostaining” shown in the 3 figures (4E, 4H and 4K ) is the same and equal to 0. Also in Fig. 5C one cannot appreciate labeling with this antibody. I suggest that this marker be excluded from the study involving xenografts and spontaneous canine mammary tumors.

In addition, the authors insist on highlighting the staining associated with areas of necrosis, which is totally nonspecific and should not be valued.

Besides, I find the histopathological characterization of the samples (xenografts and spontaneous canine mammary tumors) defective. The material was not classified according to the histopathological diagnostic systems for canine mammary neoplasia currently in use (Misdorp et al 1999, Goldschmidt et al 2011). In the revised version of the paper the authors report that the spontaneous canine mammary tumors included in the study were all solid carcinomas. However, Figure 5 reveals various acinar and tubulopapillary arrangements that characteristic of mammary carcinomas of other histologic types.

On the other hand, when describing the distribution and pattern of immunostaining in the canine neoplasms, the authors refer to “mesenchymal cells” (see text) and "myoepithelial cells" (figure5 legend), which makes no sense, assuming that those neoplasias are simple carcinomas of the solid type (thus with exclusive involvement of the luminal epithelial population, without neoplastic myoepithelial or mesenchymal component).

To study the expression of Cx26 and Cx43 in the different cell populations of the canine mammary gland (similar to that reported for human breast normal tissue - ref. 18 Monaghan and Moss 1996) it would be necessary to include samples of normal canine mammary tissue (obtained from animals free of mammary lesions), which was not done in this study.

Author Response

The following responses are to some minor points that still deserve attention:

Based on the way the authors describe the methods (2.2. Xenograft tumor model) I got the (wrong) idea that only tumors> 1cm were included in the study, and those smaller than 1cm were excluded. I suggest you to rephrase this part so that no doubt remains.

Response: Thank you for the suggestion to rephrase this description. None of the CMEC mice developed tumors. All four CMT27 mice developed rapidly growing tumors that reached the 1 cm3 cut-off goal. For the CMT12 and CF41.Mg mice, those that had no tumor development after 57 days were euthanized, but those that had slow tumor growth were euthanized at the end of the study at 77 days and these tumors ranged in size from 0.2-0.04 cm3.

The authors refer they included a statement explaining the rationale for using pan-cytokeratins and calponin markers in the M&M section (2.1. Cell lines) but I cannot find such statement.

Response: Thank you for pointing out this missing information in the process of restructuring the manuscript. The rationale has been added to the M&M section. CMEC was tested for the epithelial markers pan-cytokeratin and calponin using immunofluorescence (Figure 1B) and Western blot (Figure 1C); pancytokeratin stains luminal epithelial cells whereas calponin is a specific marker for myoepithelial breast cells. Supplemental figure has been added to support this statement. We added a statement in the result section 3.1, “CMEC cells were previously isolated from mammary gland tissues and characterized as non-cancerous epithelial cells as shown in the supplement data, Figure S1.”

The first paragraph of the discussion is not well written; some logic is missing. Rephrase please.

Response: The logic of linking neoplastic cells to the loss of cell communication was missing. Thus, we have made this linkage to the discussion.

Mammary neoplasia is the most common diagnosed tumor in bitches and thus they present a significant clinical challenge. Many biomarkers of mammary neoplasia have been reported to better understand the disease in part for early detection and prognosis. In literature, neoplastic cells have a loss of intercellular communications, particularly gap junction proteins (connexins). Here we provide for the first time that the differential pattern of connexins are presented in both non-cancerous and cancerous cells of canine mammary carcinoma. Therefore, studying gap junctions and their role in mammary cancer may provide insight into cell communication and possibly a novel therapeutic approach.

2ndparagraph (last lines) - the statement is poor and lacks bibliographic support. Rephrase please.

Response: Due to the nonselective nature of gap junctions, these channels allow small molecules, up to 1000 Daltons in size to travel from one cell to the next, providing a unique target pathway for treatment. A class of substitute quinolines, known as gap junction enhancers, can increase gap junction activity and subsequently increase efficacy of cisplatin (Ding et al 2012). Thus, restoration of gap junction intercellular communication can serve as potential pathway in combinational treatment.

Nonetheless, the major weakness of this paper lies in the histopathological and immunohistochemical study. The absence of mention to the immunohistochemical study of xenografts and spontaneous canine mammary tumors in the “Aims of the study” reinforces my conviction that this study was not delineated with this component in mind.

Response: The objective of this study is to evaluate gap junction proteins in canine mammary cells, including non-cancerous (normal) cells, and patient-derived tumors. Furthermore, primary cells as well as established cancerous cell lines were xenotransplanted to evaluate the expression of these gap junction proteins. Immunohistochemistry was an approach to evaluate the expression of gap junction proteins in addition to western blot analysis.

I am not convinced of the reliability of immunohistochemical labeling with the Cx43 antibody. The positive control used for Cx43 shows no obvious staining (Fig 4K) and therefore cannot validate the experiments performed with this marker. Take, for example, photos 4E, 4H and 4K (all related to Cx43 marking). The authors state that in Fig 4H there is no immunostaining (I totally agree) but in Figs 4E and 4K they refer to positive labelling. However, the "immunostaining” shown in the 3 figures (4E, 4H and 4K ) is the same and equal to 0. Also in Fig. 5C one cannot appreciate labeling with this antibody. I suggest that this marker be excluded from the study involving xenografts and spontaneous canine mammary tumors.

Response: In section 3.2, “with immunohistochemistry these tumors stained positive for Cx26 (Figure 4D) but there were undetectable levels of Cx43 (Figure 4E, 4I) except at necrotic areas of the tumors (Figure 4G, 4H).” In other words, 4E doesn’t have staining. In the positive control, we used heart tissue, positive stain of membranous labelling as indicated in brown.

Besides, I find the histopathological characterization of the samples (xenografts and spontaneous canine mammary tumors) defective. The material was not classified according to the histopathological diagnostic systems for canine mammary neoplasia currently in use (Misdorp et al 1999, Goldschmidt et al 2011). In the revised version of the paper the authors report that the spontaneous canine mammary tumors included in the study were all solid carcinomas. However, Figure 5 reveals various acinar and tubulopapillary arrangements that characteristic of mammary carcinomas of other histologic types.

Response: Thank you for pointing out the potential confusion of two separate studies – Figure 4 evaluating histopathological expression of connexins of xenograft tumors and Figure 5 evaluating histopathological expression of connexin of spontaneous tumors.

On the other hand, when describing the distribution and pattern of immunostaining in the canine neoplasms, the authors refer to “mesenchymal cells” (see text) and "myoepithelial cells" (figure5 legend), which makes no sense, assuming that those neoplasias are simple carcinomas of the solid type (thus with exclusive involvement of the luminal epithelial population, without neoplastic myoepithelial or mesenchymal component).

Response: Thanks for pointing the confusion between the text and figure legend. Pathologists of this manuscript have made the correction to reflect this comment.

To study the expression of Cx26 and Cx43 in the different cell populations of the canine mammary gland (similar to that reported for human breast normal tissue - ref. 18 Monaghan and Moss 1996) it would be necessary to include samples of normal canine mammary tissue (obtained from animals free of mammary lesions), which was not done in this study.

Response: We agree with your assessment; therefore, we established the canine mammary epithelial cells (CMECs) derived from normal (non-cancerous) mammary tissue. For example, Figure 2A shows the different cell populations of canine mammary gland including CMECs. We included the data of establishing CMECs in the supplement section of this manuscript.